UPDATE ARTICLE

# Further evidence for the capacity of mirror self-recognition in cleaner fish and the significance of ecologically relevant marks

Masanori Kohda[1]*, Shumpei Sogawa[1], Alex L. Jordan[2], Naoki Kubo[1], Satoshi Awata[1], Shun Satoh[1], Taiga Kobayashi[1], Akane Fujita[1], Redouan Bshary[3]

1 Laboratory of Animal Sociology, Department of Biology and Geosciences, Graduate School of Science, Osaka City University, Osaka, Japan, 2 Department of Collective Behaviour, Max Planck Institute of Animal Behaviour, Konstanz, Germany, 3 Institute of Zoology, University of Neuchâtel, Neuchâtel, Switzerland

* kohda.tanganyika@gmail.com

The Editors encourage authors to publish research updates to this article type. Please follow the link in the citation below to view any related articles.

## Abstract

An animal that tries to remove a mark from its body that is only visible when looking into a mirror displays the capacity for mirror self-recognition (MSR), which has been interpreted as evidence for self-awareness. Conservative interpretations of existing data conclude that convincing evidence for MSR is currently restricted to great apes. Here, we address proposed shortcomings of a previous study on MSR in the cleaner wrasse *Labroides dimidiatus*, by varying preexposure to mirrors and by marking individuals with different colors. We found that (1) 14/14 new individuals scraped their throat when a brown mark had been provisioned, but only in the presence of a mirror; (2) blue and green color marks did not elicit scraping; (3) intentionally injecting the mark deeper beneath the skin reliably elicited spontaneous scraping in the absence of a mirror; (4) mirror-naive individuals injected with a brown mark scraped their throat with lower probability and/or lower frequency compared to mirror-experienced individuals; (5) in contrast to the mirror images, seeing another fish with the same marking did not induce throat scraping; and (6) moving the mirror to another location did not elicit renewed aggression in mirror-experienced individuals. Taken together, these results increase our confidence that cleaner fish indeed pass the mark test, although only if it is presented in ecologically relevant contexts. Therefore, we reiterate the conclusion of the previous study that either self-awareness in animals or the validity of the mirror test needs to be revised.

## Introduction

Passing the mark test, in which subject animals touch or scrape a mark placed on their body in a location that can only be indirectly viewed in mirror, demonstrates the capacity for mirror self-recognition (MSR), which has been interpreted as evidence for self-awareness (e.g., [1–7]). Variations of this test have been applied to many species of vertebrates. Most often, the results are clearly negative, including studies on lesser apes, monkeys, pig, dog, cat, panda, crows, and parrots (e.g., [1–2,5,8–16]). However, a small number of socially intelligent species including

**Data Availability Statement:** All relevant data are within the paper and its Supporting Information files.

**Funding:** This study was financially supported by KAKENHI Grants from JSPS to MK (nos. 17K18712, 19F19713, 19H03306 and 20K20630) and to SS (nos. 20J01170) and by the Osaka City University Strategic Research Grant 2018 and 2019 for Top Priority Researches to MK. RB is supported by the Swiss Science Foundation, Grant 310030_192673 / 1. ALJ is supported by the Max Planck Society and Deutsche Forschungsgemeinschaft Cluster of Excellence 2117 "Centre for the Advanced Study of Collective Behavior" Grant 422037984. The funders had no role in study design, data collection and analysis, decision to publish, or preparation of the manuscript.

**Competing interests:** The authors have declared that no competing interests exist.

**Abbreviations:** GLMM, generalized linear mixed model; LMM, linear mixed model; MSR, mirror self-recognition; SEM, standard error of the mean; TL, total length; VIE, visible implant elastomer.

elephant, dolphin, horse, magpie, and a crow have been argued to have passed the test (e.g., [17–23]), although this interpretation has been criticized [7,13]. As a consequence, only the evidence on chimpanzees, and to a lesser degree on orangutans, has so far obtained unequivocal approval as evidence for MSR and hence self-awareness [7]. Furthermore, there is disagreement on whether there are intermediate levels of self-awareness [24,25] or whether self-awareness may represent a cognitive discontinuity [7,8].

Morgan's canon states that simpler explanations must be excluded in order to more complex cognitive interpretations being acceptable, and adhering to this logic suggests that we must acknowledge the concerns regarding evidence for MSR in nonapes raised by Gallup and Anderson [7]. The authors propose the following criteria that need to be fulfilled in combination to conclude that a species passed the mark test: (i) repeated studies (ideally by different laboratories) showing positive results; (ii) linked to the first point, a reasonable number of individuals should pass the test; and (iii) additional experiments should exclude alternative explanations for mirror-related actions. These additional experiments can be fairly simple but telling. For example, 2 rhesus macaques that had shown habituation to a permanent mirror (without showing evidence for MSR) started to behave aggressively again after the mirror was simply moved to another side of the cage, clearly showing that they did not recognize themselves [26]. Also, individuals naive to mirrors should not pass the mark test spontaneously, as was the case for the 2 mirror-naive chimpanzees in the original study, which did not inspect their mark within the first 30 minutes of exposure [1]. Additionally, marked individuals that see another marked individual rather than their own mirror image should not attempt to remove the mark. Finally, both Gallup and Anderson [7] as well as de Waal [24] emphasize that the interpretation of results becomes less clear if the mark is not just painted on the skin but attached below the skin. This is because the physical sensation of the mark (or head implant for physiological studies), together with seeing the mark in the mirror, may trigger mark-related behavior and then also inspection of other body parts, as, for example, in rhesus macaques [27], a species that otherwise fails the mark test. Thus, it was argued that the monkeys may have learned contingencies rather than recognizing themselves in the mirror [24]. Note, however, that this interpretation differs from the one by the authors, who propose that the salience of the mark triggered closer inspection and, as a consequence, MSR [27].

Marking procedures are also a crucial element of the debate triggered by recent results on the behavior of cleaner fish *Labroides dimidiatus* when exposed to a mirror [25]. Cleaner wrasse obtain their food by eating parasites and mucus of the surface of other fish, so-called clients [28], and Gnathiid isopods are their main food [29]. These small crustaceans appear as small dark dots on clients. Kohda and colleagues. [25] used this feature of cleaner fish ecology to mark subjects in a salient way, i.e., by injecting a brown elastomer marking on the throat such that the marking was only visible when subjects swam upward in front of a mirror. In this previous experiment, brown marking, but not invisible elastomer implants, caused 3 out of 4 mirror-experienced cleaners to scrape their throats several times after swimming in front of the mirror, but not when a mirror was absent [25]. Despite these results, a number of criticisms by de Waal [24] and Gallup and Anderson [7] potentially apply, including the possibility that the elastomer marks also produced some physical sensation, akin to the head implant in rhesus macaques.

Here, we provide results on various follow-up experiments designed to challenge the interpretation by Kohda and colleagues [25] that cleaners pass the mark test. The first aim was to test whether the earlier results can be reproduced by a new experimenter with a larger sample size. Furthermore, we tested whether the brown color of the marking was crucial for the throat-scraping responses. If cleaners are only responding to ecologically relevant markings,

then blue or green marks would not elicit throat scraping. In order to obtain further information on the role of cleaners feeling the mark, we injected the mark deeper in some cleaners in a further experiment. If such marking caused irritation such that cleaners scraped their throat in the absence of a mirror, but did not do so under normal (shallow) marking procedures, it would suggest that the standard marking is not comparable with a skull implant for electrophysiological experiments in rhesus macaques, but that deeper marking may be. Furthermore, we marked mirror-naive individuals and exposed them to a mirror. Based on the 2 data points on chimpanzees [1], we expected that these subjects should show no or at least less throat scraping during the 120-minute exposure. Introducing a new experimental paradigm, we also marked mirror-experienced cleaner pairs that could see each other through a transparent barrier. If seeing any cleaner with a brown mark on the throat would somehow remind subjects of the mark on their own throat, then we would expect throat scraping in this experiment as well. Our final experiment involved changing the position of the mirror once cleaners had stopped to aggress their own reflection. If moving the mirror would lead to renewed aggression, this result would emphasize the importance of learned spatial contingencies as opposed to self-recognition.

Before passing mark test, animals progress through Stages 1 to 3 (chimpanzee, dolphin, elephant, and cleaner fish [1,17,18,25]). The first stage involves social interactions including aggression (animals mistake a reflection for another conspecific), the second stage involves repetitive atypical behavior against mirror reflection, by which animals are thought to check the contingency in movements between their own body and the mirror reflection (i.e., testing whether the reflection is self or not), and the final stage when animals perform self-directed behavior using the mirror (understanding the reflection to be self). These processes are considered to be a basic evidence for MSR [1,17], and in this study, we also describe these behaviors in cleaner fish.

## Results and discussion

### Replication of mark test using 8 new fish (Experiment 1)

Initially, the 8 test fish did not scrape their throats during a 2-hour period of no treatment with a mirror (Fig 2A, Table 1). None of these 8 fish that were given the sham mark on the throat scraped their throats. Injecting a brown mark that resembled a parasite on the throat (Fig 1A) did not elicit scraping the throat when the whole mirror was covered with white screen, but all subjects scraped the mark during a 2-hour period in which the mirror was exposed (Fig 2A, Table 1). The mean scraping frequency of the throat was 2.31 ± 0.27 times standard error of the mean (SEM)/h ($n = 8$), which was not different from that of the previous study ([25]: 3.11 ± 1.26/h, $n = 4$, see S1 Data for raw data, Mann–Whitney U test, U = 11.00, $p = 0.44$). When color marked, subject fish also spent significantly more time close to the mirror in a posture that would allow them to see throat mark in the mirror compared to earlier periods when they were still nontreated and treated with sham mark (Fig 2B, Table 1). Furthermore, the swimming speeds were slightly higher during the actual mark test than in the period of no mark but also with mirror, while swimming speeds were intermediate in the other 2 conditions (Table 1).

Subjects selectively increased the scraping of their throat; they scraped other body parts in all 4 periods at rather constant frequencies (Table 1). The results suggest that these body rubbings are rather independent of the throat scraping induced by the color mark with mirror. Frequencies of other actions such as fin election and touching mirror by mouth were not different among the treatments (Table 1). Taken together, throat scraping is not a by-product of a

**Table 1. Mean ± SEM cleaner fish behaviors in 4 different treatments (*n* = 8 individuals) and results of statistical tests aimed at detecting differences between treatments.** See S1 Data for raw data.

| Behavior | No treatment with mirror | Sham mark with mirror | Brown mark no mirror | Brown mark with mirror | Statistics* $\chi^2$ | df | *p* |
|---|---|---|---|---|---|---|---|
| Frequency of throat scraping (times/h) | 0.00 ± 0.00[a] | 0.00 ± 0.00[a] | 0.00 ± 0.00[a] | 2.31 ± 0.27[b] | 24.00 | 3 | <0.0001 |
| Time in posture reflecting the throat (sec/10 minutes) | 24.25 ± 2.42[a] | 25.63 ± 3.63[a] | – | 73.50 ± 7.19[b] | 12.00 | 2 | 0.002 |
| Swimming speed (mm/sec) | 94.35 ± 8.27[a] | 97.80 ± 8.89[a,b] | 96.45 ± 8.71[a,b] | 100.05 ± 9.46[b] | 11.22 | 3 | 0.01 |
| Frequency of body and face scraping (times/h) | 3.56 ± 0.38 | 3.50 ± 0.33 | 3.69 ± 0.33 | 3.75 ± 0.31 | 0.30 | 3 | 0.96 |
| Frequency of fin spreading (times/10 minutes) | 6.25 ± 0.86 | 6.13 ± 0.88 | 6.50 ± 0.71 | 7.13 ± 0.72 | 0.30 | 3 | 0.96 |
| Frequency of touching mirror by mouth (times/10 minutes) | 4.13 ± 0.79 | 4.25 ± 0.90 | – | 4.38 ± 0.78 | 0.00 | 2 | >0.99 |

*A LMM was applied to swimming speed of the subject fish (*n* = 5 times measurements per treatment per individual) and Friedman tests to the other 5 behavioral factors. Different letters (a vs. b) denote statistically significant differences by multiple comparisons using Tukey contrasts (swimming speed) and exact Wilcoxon signed-rank tests with sequential Bonferroni correction (frequency of throat scraping and time in posture reflecting the throat).
LMM, linear mixed model; SEM, standard error of the mean.

general change in activity patterns but is evoked by the motivation to remove the "harmful" color mark resembling a parasite.

These new results, obtained by a new independent generation of students, strongly increase our confidence that throat scraping behavior is a common and selective response of cleaner wrasse in the mark test rather than the behavior of few exceptional individuals. Moreover, all 6 fish used in Experiment 2 (see "Green and blue marks do not elicit scraping (Experiment 2)") scraped the brown mark in the presence but not in the absence of a mirror. Thus, all 14 fish subjected to the standard test passed in the present study. In total, this brings the number of

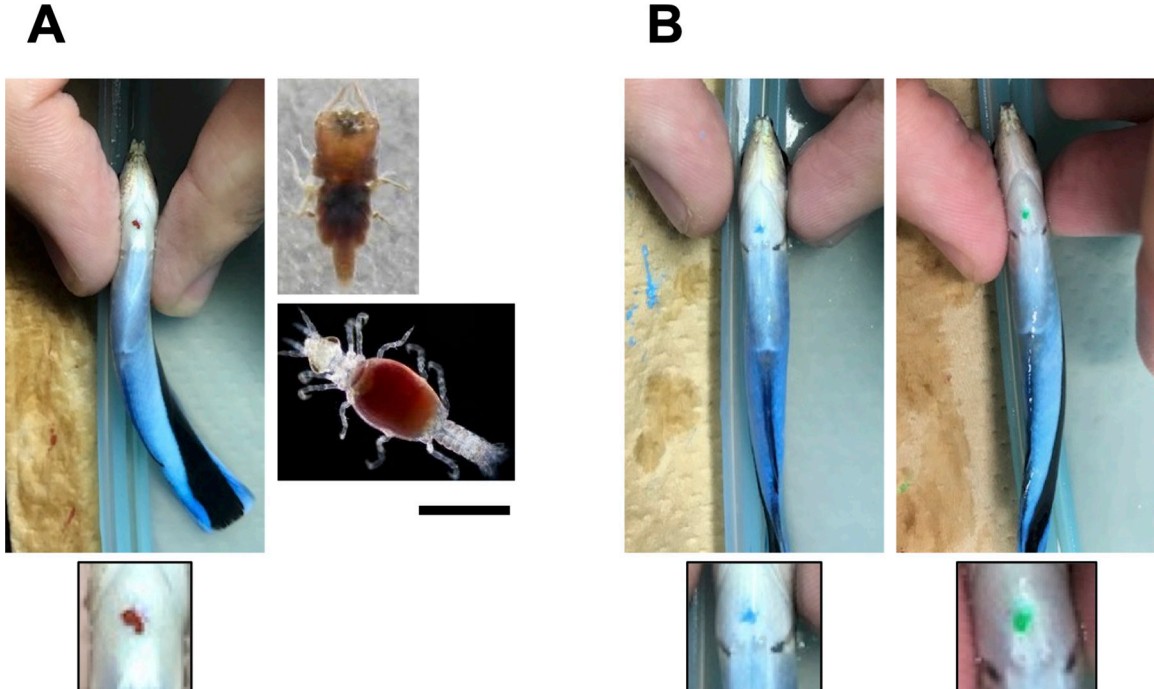

**Fig 1.** Ecologically relevant color mark (brown) on the throat of cleaner fish and ectoparasite (sea stag) (**A**). Bar is 1 mm. Meaningless color mark (green and blue) on the throat (**B**). These cleaner fish are just after mark injection and are still in the anesthetized condition.

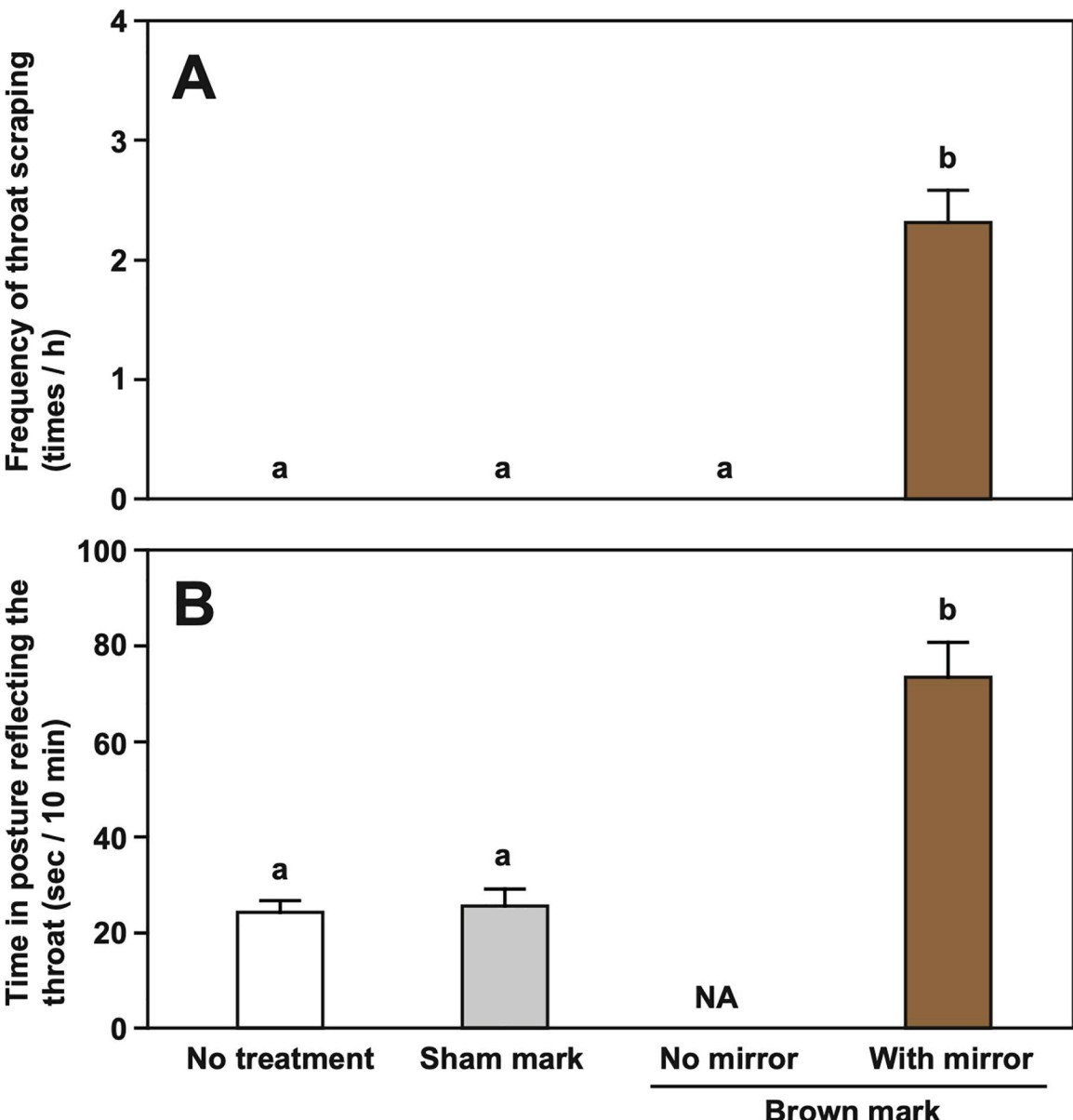

**Fig 2.** Mean frequency (± SEM/h) of throat scraping on the substrate **(A)**, and mean time (sec ± SEM/10 minutes) of position reflecting their throat on a mirror **(B)** of the 8 cleaner fish *L. dimidiatus* in Experiment 1: during the periods of no treatment, sham mark, brown mark without mirror, and brown mark with mirror presentation. Different letters denote statistically significant differences by exact Wilcoxon signed-rank tests with sequential Bonferroni correction. See S1 Data for raw data.

cleaner wrasse tested to 18 (4 fish from Kohda and colleagues [25]), the largest sample size for any nonhuman species tested for MSR capacity outside of chimpanzees [13]. Cleaner wrasse also currently shows the highest rate of passing with this large sample size (94% = 17/18; and one failing fish in Kohda and colleagues [25]). In contrast, only a small proportion of individuals pass the test in the animal species of MSR capacity reported hitherto, e.g., ca. 40% in chimpanzee ($n = 97$), 50% in orangutans ($n = 6$), 30% in gorillas ($n = 15$), 30% in Asian elephant ($n = 3$), and 40% in magpie ($n = 5$) [20,30–33], except 100% in dolphin ($n = 4$) [17,22].

## Green and blue marks do not elicit scraping (Experiment 2)

Kohda and colleagues [25] proposed that the high pass rate by cleaner fish could be due to the mark visually resembling a parasite, making the aim of its removal an ecologically relevant task. In contrast, other species will not have such a motivation because of a simple mark, and indeed chimpanzees soon lose interest [30]. Kohda and colleagues [25] interpreted the lack of cleaners responding to the transparent sham mark as evidence that cleaners do not feel a physical stimulus. In contrast, Gallup and Anderson [7] as well as de Waal [24] argued that the transparent sham mark may be different enough so that cleaners do not feel its presence. More generally, the fish may need to feel a physical stimulus simultaneously with a visual stimulus to perform mark removal behaviors [7,24]. To distinguish between the alternative explanations, we used 6 cleaners to subsequently inject green, blue, and brown marks (Fig 1), counterbalancing the order between subjects. Neither blue nor green marks induced throat scraping in the presence of the mirror, while all subjects scraped their throats when injected with the brown mark (Poisson generalized linear mixed model [GLMM], $\chi_3^2 = 74.78$, $p < 0.0001$; Fig 3A). Furthermore, when injected with either blue or green mark, subjects infrequently assumed a posture reflecting the throat mark, i.e., not more frequently than in the control with no injection and significantly less than when injected with a brown mark (linear mixed model [LMM], $\chi_3^2 = 23.34$, $p < 0.0001$; Fig 3B). The scraping frequency of 2.42/h ± 0.55 SEM ($n = 6$) was not different from that in Experiment 1 (Mann–Whitney U test, U = 23.50, $p = 0.99$). Note that no fish scraped the green, blue, and brown marks during 2-hour observation periods in the absence of a mirror.

This experiment demonstrates that visual information—with or without a potential physical stimulus from the injection—is not enough to elicit throat scraping. Instead, it appears that the visual stimulus needs to be salient and of negative valence. Only an ecologically relevant mark suggesting the presence of an ectoparasite-induced throat scraping.

## Placing the mark deeper into the fish tissue (Experiment 3)

In this additional experiment, we aimed at testing how fish would respond to a physical stimulus in their throat. The same amount of elastomer was injected ca. 3 mm (rather than the standard < 1 mm) from the outer layer of skin into the throat of 6 new fish. The deep mark was hardly visible, apart from a small brown dot at the injection point. The fish were observed before and during mirror exposure. They scraped their throat regardless of whether a mirror was absent or present and at similar rates (mirror absent: 2.50 times ± 0.45 SEM/h; mirror present: 2.75 ± 0.76 SEM/h; $n = 6$, exact Wilcoxon signed-rank test, V = 4.00, $p = 0.75$). The result shows that a painful or itching mark does not require a mirror to elicit self-scratching. In contrast, summing up data of the current study's Experiment 1 and 2, a total 14 different cleaners never scratched their throat with the standard marking procedure in the absence of a mirror, over extended periods of either 28 hours (brown mark) or 24 hours (green and blue marks). We therefore conclude that the standard mark is unlikely to be perceived as painful or itching and that any throat scratching in the presence of a mirror is only due to the visual signal resembling an ectoparasite.

## Behavior of marked mirror-naive individuals (Experiment 4)

In the original mark test experiments on chimpanzees, mirror-trained, but not the 2 mirror-naive individuals, scratched at the mark within 30 minutes of exposure [1]. We found qualitatively similar results in a replication of this treatment, exposing 9 mirror-naive marked individuals for 2 hours to a mirror (see S1 Data for raw data). The mean scraping frequency of these individuals was 0.44 ± 0.18 SEM/h ($n = 9$), significantly lower than the mark test with

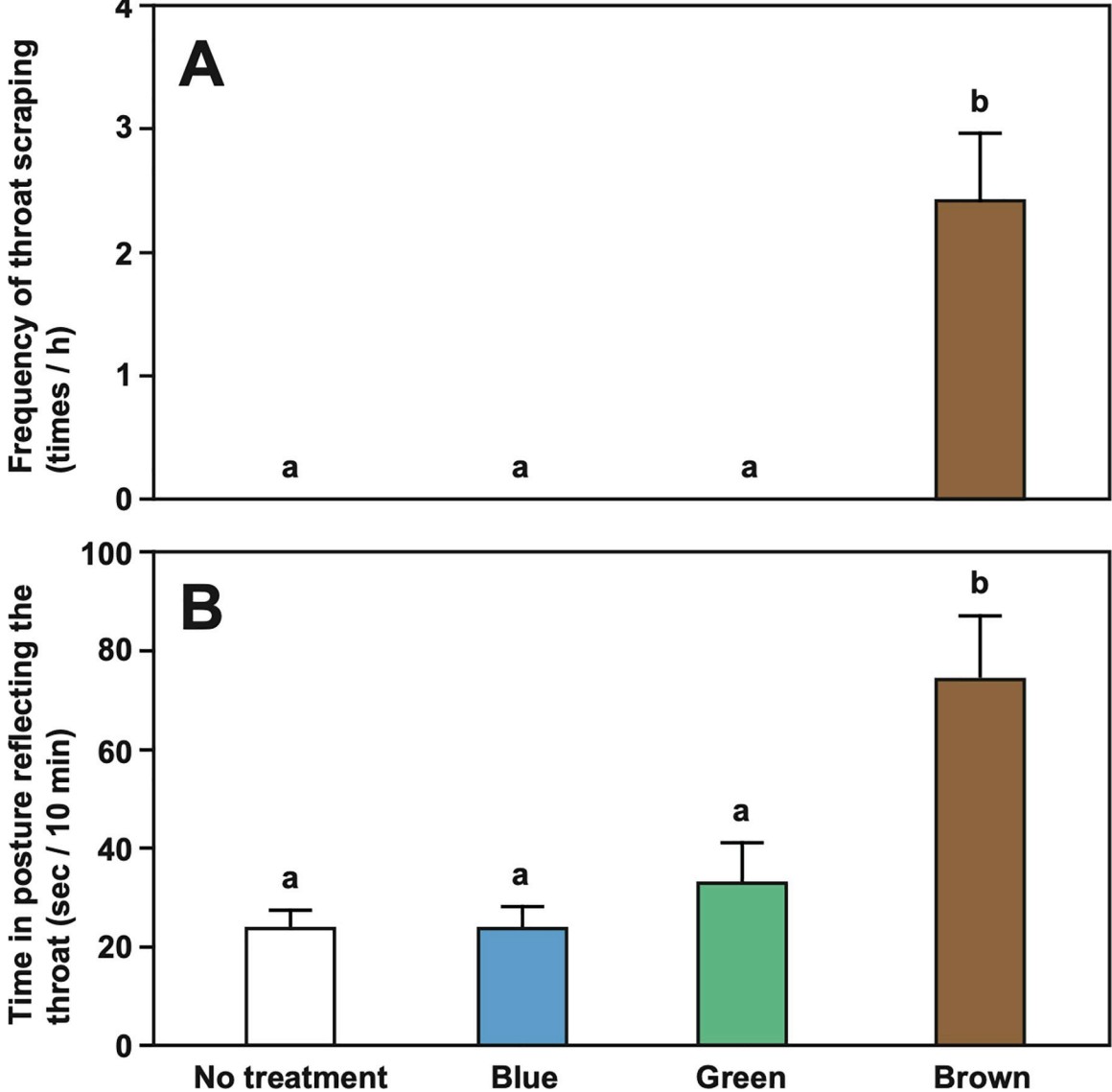

**Fig 3.** Mean frequency (± SEM/h) of throat scraping on the substrate (**A**) and mean time (sec ± SEM/10 minutes) of position reflecting their throat on a mirror (**B**) in no treatment, blue mark, green mark, and brown mark by cleaner fish *L. dimidiatus* in presence of mirror. No throat scrapings were observed in all cases in the absence of mirror, and the results are omitted. See S1 Data for raw data. SEM, standard error of the mean.

brown mark in the original type, 2.36 ± 0.27/h ($n$ = 14) (Mann–Whitney U test, U = 4.50, $p < 0.0001$). Nevertheless, we note that 5 out of 9 mirror-naive fish scraped their throat within 2 hours, although their scaping frequency was still lower than that of mirror-experienced subjects (0.80 ± 0.20 SEM; $n$ = 5, Mann–Whitney U test, U = 4.50, $p = 0.002$), partly because mirror-naive individuals tended to start scraping later (mirror-naive 82.11 ± 32.28 minutes SEM versus mirror experienced 39.60 ± 8.48 minutes).

Gallup and Anderson [7] hypothesized that mirror-naive individuals should not be or be less responsive to a marking because they do not recognize the reflection as self. While our results fit this hypothesis qualitatively, we anticipate that skeptical colleagues will view them as evidence that throat scraping in cleaners in the presence of a mirror is not evidence for self-

recognition. It would be difficult to assume that the speed of learning the mirror contingencies seemed be enhanced by the subjects seeing the mark in the mirror. To consider the cause fairly, we need quantitative data of mirror-related behavior of nonmarked naive fish in the initial 1 or 2 hours after mirror setting, but not have it. Importantly, however, we have no benchmark data from other species for comparison in order to assess whether the marked cleaner fish learned the mirror contingencies atypically fast. As it stands, Gallup [1] tested only 2 mirror-naive individuals during 30 minutes of exposure. Our fish were exposed for 120 minutes, and the confines of the aquaria (45-cm length) as well as the almost 360-degree vision of cleaners ensured almost permanent exposure. Therefore, conducting more experiments on mirror-naive chimpanzees (as well as other mirror-naive animals of MSR capacity) is necessary to test to what extent a visual marking affects the process of passing the mirror test in the species that shows the most unambiguous evidence for MSR.

### Behavior of marked mirror-experienced individuals paired with another marked individual (Experiment 5)

We conducted this experiment on 6 individuals that had already passed the mirror test, as an additional way to test whether feedback between a visual and a physical stimulus may cause throat scraping. We placed 2 subjects in adjacent aquaria separated by transparent glass. Subjects showed largely reduced aggressive behavior toward each other within 2 to 3 days, similar to the previous study (see Fig 1C in [25]). Both fish were marked in the standard way at night after the fourth day. None of the 6 subjects ever scraped its throat during 120 minutes when exposed to each other in the next morning (see S1 Data for raw data).

This result shows that a visual ecologically relevant stimulus on another fish is not enough to induce throat scraping in marked subjects. Instead, subjects need to see the marking in their mirror image, and by extension that contingency between own movement and that of the mirror image is crucial for subjects to meet criteria for passing the mirror test.

### Fish responses to moving the mirror (Experiment 6)

This experiment is actually the simplest control proposed by Gallup and Anderson [7] to challenge the notion of MSR in any animals that stopped behaving aggressively toward its mirror image. If moving the mirror reignites aggressive behavior, the animal has only learned a spatial contingency. We transferred 6 mirror-trained cleaner fish to a new tank, where they were first exposed to a mirror on one side end, and after 3 days to a mirror on the other side end. Subjects ($n = 6$) did not show any aggression toward their mirror image during the 120 minutes of exposures to the first and then the second mirror (see S1 Data for raw data).

The results show that cleaners do not learn a spatial contingency that allows them to eventually stop aggressing their mirror image. Instead, cleaners must learn about their own individual features and selectively stop showing aggression toward these features, independently of where they see them. Note, however, that the lack of aggression by itself does not show that cleaners recognize the mirror image as self. Alternatively, the subjects could have habituated to what they perceive as one specific other individual, and hence show no aggression no matter where they encounter it (e.g., [34]). Nevertheless, the cleaners clearly outperform rhesus macaques in this particular experiment, as the latter were highly sensitive to the movement of the mirror [35].

### General discussion

There is a current controversy regarding the interpretation of results from the mirror task, reignited by recent results on cleaner fish [7,24,25]. Our aim was to present cleaner fish to a

variety of largely new experiments aimed at challenging the interpretation by Kohda and colleagues [25] that cleaner fish show self-recognition. We welcome a general discussion on these new results. As we see it, the additional experiments largely support the notion that cleaner fish indeed show self-recognition in the mirror task. We have greatly increased sample size, showing that throat scraping is a general behavior of cleaners when marked and exposed to a mirror. Subjects need to see on their mirror image rather than the mark on another individual to scrape their own throat. Furthermore, cleaners recognize the individual in the mirror rather than having learned that a fish at a certain location should no longer be confronted. In each of these experiments, cleaners could have behaved in ways that would have invalidated the conclusion by Kohda and colleagues [25] that cleaners pass the mirror test, but they never did. We acknowledge that the cleaners' behavior in each single additional experiment can probably be explained without invoking MSR. However, the combined accumulated evidence should be more difficult to be dismissed than the previous study.

A remaining potential shortcoming of the current study is that the mark was injected rather than painted. We do not see how this can be changed in fish. We showed that a deeper injection causes scraping frequencies that are independent of the presence/absence of a mirror. Furthermore, the results from the 2 experiments with marked fish suggest that the visual input must be the mirror image rather than a marked conspecific. Both results make it less likely that the standard marking procedure causes a visual sensory feedback loop during mirror exposure, as proposed by de Waal [24]. At the minimum, cleaner fish are able to learn that only the mirror image provides contingencies for self and use this knowledge to scrape a body part with an apparent parasite attached when spotted in the mirror.

Mirror-naive cleaners scraped their throat less frequently than mirror-experienced individuals or even not at all. Thus, one could argue that the results from this experiment qualitatively fit evidence for cleaner self-recognition. On the other hand, the timing with which several mirror-naive cleaners started using the mirror reflection to scrape their throat was early. Thus, the data could currently be used as evidence both for or against MSR [7]. To solve the puzzle fairly, we need intensively observe behaviors of nonmarked naive fish in initial hour of mirror presentation. We consider any strong conclusions of the strange timing of this fish premature, as we also lack quantitative data on other species. In his classic study, Gallup [1] had only tested 2 naive chimpanzees for 30 minutes, and no similar data have been collected in other studies as far as we are aware.

One important conclusion from our study is that cleaners only respond to markings of apparently high ecological relevance. We therefore encourage colleagues to think hard about which marks could be relevant for their study species in order to increase the likelihood of responses [36]. Only particularly curious and/or playful species may inspect any marking, regardless of its ecological relevance. Fish are generally not known for curiosity and playfulness (but see [37]), making ecological relevance of the mark a potentially imperative prerequisite. This also implies that we cannot expect a fish showing inspection behavior of otherwise invisible body parts when in front of a mirror, and fish would not be able to touch these parts anyway. Our subjects were all wild caught as adults and hence had plenty of experience searching and eating small crustacean ectoparasites prior to our experiments. Unfortunately, the life cycle of cleaner wrasse cannot be completed in the laboratory, preventing experiments on parasite-naive individuals to test whether the behavior of current subjects was a response to an innate or acquired stimulus. Independently of the answer to that question, cleaners have to perceive the mirror image as relevant for self, including parts they had never seen without a mirror.

In conclusion, we propose that the validity and/or the conclusions from the mirror task need further investigation. Given the negative results on a great variety of large-brained

endotherm vertebrates, the positive results for cleaner fish present a puzzle. High ecological relevance of the stimulus, in combination with experience with the stimulus, may potentially be an important part of the answer. Familiarity with all components of the task may greatly enhance the probability that subjects are able to combine the available information to form new insights. We note that cleaner fish show evidence for a variety of unexpected advanced cognitive abilities. For example, cleaner wrasse use predators as social tools against aggressive clients [38] and can generalize across predatory species in learning experiments mimicking the social tool use scenario [39]. Furthermore, cleaners apparently use configurational learning to give priority to ephemeral clients over more permanently accessible ones [40–42]. Finally, cleaners can incorporate what other cleaners can or cannot see [43], a supposedly key building block of a theory of mind [44]. Cleaner wrasse have an average brain to body ratio for a labrid fish [45], making it likely that their 2,000 interactions with client reef fish provide such abundant learning opportunities that cleaners eventually reach more advanced insights within the narrow ecologically relevant context. With this perspective, more species may be found to show evidence for MSR if the task can be made ecologically relevant to them.

The main open question in our view is how MSR relates to self-awareness. We cannot provide an answer. Nevertheless, we agree with de Waal [24] that self-awareness is not necessarily an all or nothing. Indeed, any moving animal must have a basic notion of self, i.e., the size and shape of its body, in order to avoid bumping into obstacles [46]. In contrast, recognizing oneself in a mirror does not necessarily imply the presence of other, supposedly advanced cognitive processes. Children recognize themselves in a mirror long before they pass the Sally–Anne test for conscious attribution of intentions and beliefs to other individuals [47,48]. Conversely, specific brain damage prevents MSR without impairing theory of mind in adult humans [4]. Similarly, MSR may need to be combined with mental time traveling abilities to grasp the concept of death. Given the available evidence, we conclude that the degree of self-awareness may well differ between species and in ways that are independent of performance in the mirror test.

## Materials and methods

### Ethics statement

All experiments were conducted in compliance with the animal welfare guidance of the Japan Ethological Society and were specifically approved by the Animal Care and Use Committee of Osaka City University.

### Subject animals and housing

The cleaner wrasse *L. dimidiatus* inhabit coral reefs and rocky sea shore in tropical and subtropical areas in the world, and take ectoparasites of client fish ([49], see Fig 1B). This is a small fish, up to 15 cm in total length (TL), of protogynous hermaphrodite, changing sex from female to male, and they have harem polygynous mating system [50,51]. This fish is a model species for the study of fish cognition [52–54], and many aspects of fish social cognitive capacities have been reported from this fish, for example, the strategic use of tactical deception [55], transitive inference [56], a strong ability to delay gratification [57], a base for theory of mind [43], and MSR [25].

This study was conducted at the Laboratory of Animal Sociology, Department of Biology and Geosciences, Graduate School of Science, Osaka City University, Japan [25,34,58,59]. We used a total of new 35 wild-captured cleaner fish via commercial ornament fish shops. Fish were between 60 and 76 mm in TL, and individuals of this size are functionally female [25]. Fish were housed in separate tanks (45 cm × 30 cm × 28 cm), and each individual was kept for at least 1 week to be acclimated to captivity prior to the start of the experiments. Fish were kept

in a 12:12 hour of light:dark cycle throughout the study. Almost all experiments were conducted in the subjects' home tank, with the exception of Experiment 6. Each tank contained a PVC pipe for fish sleeping shelter and a rocky block (10 cm × 5 cm × 5 cm) as a potential body scraping site. Coral sand and coral pebble formed a 2- to 3-cm thick substrate on the tanks' bottom. The water was aerated and filtered, and temperature was kept between 25 and 26°C. Cleaners were fed a small piece of diced fresh shrimp meat every day. These tank conditions were the same as in the previous study [25].

We attached a 45 × 28 cm² high-quality mirror on one glass wall inside the tank, which was initially completely covered with a white plastic sheet (45 × 28 cm²). The methods of mirror presentation to subject fish were the same as those employed in Kohda and colleagues [25]. That is, at the start of the MSR test, the white sheet on the mirror was removed, and thereafter the subject fish was exposed to the mirror until the end of the series of experiments, with the exception of a several hours experiment during which the mirror was completely covered with the white sheet (see below). Fish scraped the throat many times at the block set at the corner of the mirror as in the previous study [25].

### Description of fish behavior before mark test

Kohda and colleagues [25] describes 3 stages from cleaners' interacting with its mirror reflection, each phase is characterized by typical behaviors: socially aggressive behavior of mainly mouth–mouth fighting in Stage 1 (in the first 3 days after mirror presentation), unusual behavior against mirror reflection in Stage 2 (third to fifth day), and watching their reflection frequently close to mirror (after 5 days). The former 2 stages almost finish within the fifth day of mirror presentation, and then the last stage of watching reflection starts. Stage 2 will be time for contingency testing of movement and Stage 3 will be time of self-directed behaviors. These 3 stages will be consistent with other MSR animals [1,17]. The last stage is indicative of subjects having recognized the mirror reflection as themselves and hence being ready to pass the mark test [25].

In the present study, these social reactions, mouth–mouth fighting, atypical contingency testing behavior, and watching reflection were observed and are shown in Fig 4 with description of contingency testing behaviors in Table 2. The video data were taken by Fujita and Sogawa in relation to the replication of mark test of 8 fish in Experiment 1 (see the procedure of Experiment 1 later mentioned) and analyzed by Kubo. Fig 4 indicates mouth–mouth fighting occurred in the first 3 days (Stage 1) and atypical behaviors against mirror (Table 2) with the peak of third to fifth day (Stage 2) and the self-directed behaviors of watching its face or body within 5 cm from mirror would start from fourth day (Stage 3). Atypical behaviors categorized into 5 types, which could be regarded as contingency testing behaviors [25]. Thus, the 3 stages were largely similar to the previous study [25], and we could start to test subjects after 1 week of mirror presentation. Self-directed behavior observed were only subjects watching their mirror reflection of body or face from near the mirror as in the previous study.

### Procedure of provisioning mark

As in the previous study, visible implant elastomer (VIE) marking (Northwest Marine Technology, Shaw Island, United States of America) via subcutaneous injection on throat was used [25]. Fish cannot see the marks on their throat directly (Fig 1). VIE marking was made as follows: Fish sleeping in the PVC-pipe shelter were taken out from their tanks at night together with shelter, and they were placed in eugenol solution to achieve mild anesthesia (using FA100, Tanabe Pharmacy, Japan). Then, the color mark was injected subcutaneously on the throat of the subject fish (Fig 1). The "standard" VIE marking (i.e., brown) provided a color

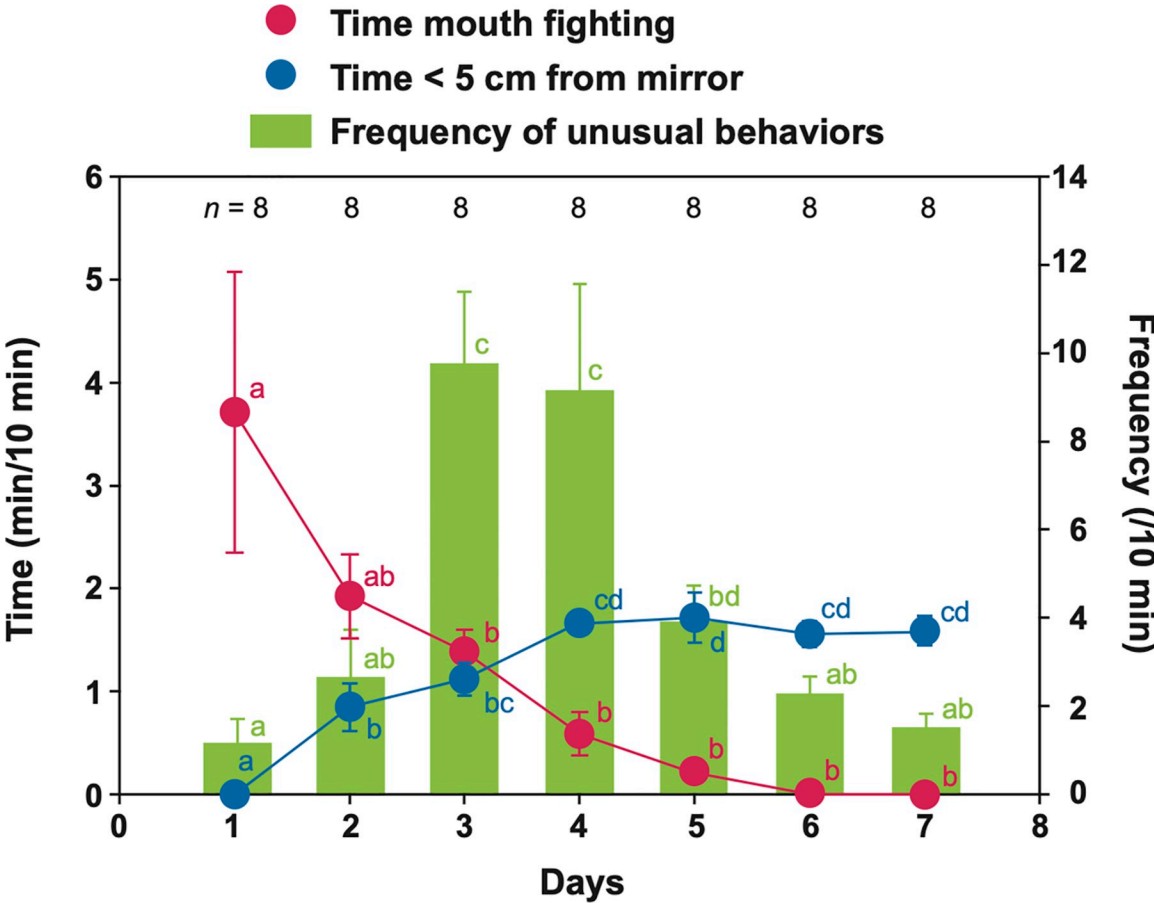

**Fig 4. Changes in social responses of cleaner wrasse toward the mirror during the first week.** Mean ± SEM for the time spent mouth–mouth fighting (red), time of watching reflection within 5 cm of the mirror (blue), and frequency of mirror testing behavior/10 minutes (green) (see Table 2). Time spent mouth fighting: LMM, $\chi_6^2 = 31.07$, $p < 0.0001$; Time < 5 cm from mirror: LMM, $\chi_6^2 = 63.38$, $p < 0.0001$; and frequency of unusual behaviors: negative binomial GLMM, $\chi_6^2 = 59.42$, $p < 0.0001$. Different letters of the same colors denote statistically significant differences by multiple comparisons using Tukey contrasts. See S1 Data for raw data. GLMM, generalized linear mixed model; LMM, linear mixed model.

**Table 2. Total occurrence of contingency testing behaviors shown by 8 fish during 20-minute observation/day in the first 7 days after mirror presentation.**

| Fish code | Behavior 1 | Behavior 2 | Behavior 3 | Behavior 4 | Behavior 5 | Total |
|---|---|---|---|---|---|---|
| #1 | 5 | **10** | 8 | 4 | 2 | 29 |
| #2 | **34** | 2 | 0 | 17 | 4 | 57 |
| #3 | 1 | 1 | 1 | **10** | 3 | 16 |
| #4 | 1 | 1 | 0 | **3** | 0 | 5 |
| #5 | 6 | 7 | 2 | **20** | 3 | 38 |
| #6 | 9 | **28** | 3 | 4 | 4 | 48 |
| #7 | 5 | 0 | 3 | **17** | 4 | 29 |
| #8 | 5 | 3 | 5 | **7** | 0 | 20 |
| Total | 66 | 52 | 22 | 82 | 20 | 242 |

Bold numbers show the most frequently observed pattern in each fish.

Behavior 1: Rapid dashing along the mirror surface for 10 to 30 cm; Behavior 2: Fish spreading all of their fins and quickly quivering the body for ca. 1 second at a distance 5 to 10 cm from the mirror; Behavior 3: Fish rapidly dashing toward the mirror but stopping before crashing into it; Behavior 4: Body shaking while looking at mirror; and Behavior 5: Face shaking while looking at mirror. Each behavior occurs within 1 second.

dot that looked like an ectoparasite in the wild. After VIE marking, fish were returned to their home tank, and the mirror in their tank was covered with a white board. In the next morning, fish behavior was recorded by a video camera (Sony Camcorder HD RCX-680, Tokyo, Japan), and behavioral analysis was conducted using these video recordings.

**Experiment 1: Replication of mark test.** Eight fish were used in this replication of the original mark test by Kohda and colleagues [25] in Experiment 1. During the first 5 days, fish initially exhibited aggression and contingency testing behaviors, but these decreased and were infrequently observed by 7 days (Fig 4) as in the previous paper [25]. On the eighth day of continuous mirror presentation, fish behavior was recorded for 2 hours before any treatments of marking. In the following night, we sham marked subjects by VIE marking with a transparent nonpigment gel subcutaneously on the throat. From 9 AM in the next morning, fish behavior was recorded for 2 hours with the mirror uncovered. Two nights later, VIE marking with brown color was done, and fish behaviors were recorded for 2 hours in the morning of the next day while the mirror was covered with the white plastic sheet (color mark without mirror). After this observation, the mirror was uncovered, and their behavior was recorded for another 2 hours (color mark with mirror). As found in other studies [60–62], this marking procedure did not alter fish behavior, and fish swam normally in the morning following the injection as they had done during the previous study [25]. The brown color mark was injected right next to the transparent mark (Fig 1A). Even with both marks, the total volume of the tag was smaller than the minimum recommended amount for tagging even for small fish. In Experiment 1, all of VIE marking, video recording, and video analyses were conducted independently by new members of Fujita, Kubo, and Sogawa, and the members of the previous study teams did not conduct the experiments [25].

In Experiment 1, Fujita and Sogawa observed and counted the number of throat scraping behaviors in video recordings during 2 hours in no treatment, sham mark, color mark in the absence of mirror, and color mark in the presence of mirror. After their recording, the videos were checked by Kohda, who independently identified the same throat scraping events, which were all identified as mark scraping. We also quantified fish posturing behavior against mirror. Cleaners have the best view on their throat if they swim up vertically in front of the mirror. We hence quantified the time spent in a vertical posture toward the mirror within 5 cm from it during each exposure to the mirror. Kubo analyzed aggression duration (seconds) against mirror, number of atypical behavior in front of mirror, i.e., contingency testing behavior [25], and duration (seconds) of watching its mirror reflection, i.e., self-directed behavior [25] for 20 minutes a day in the first week after mirror presentation appearing in Fig 4. Kubo also analyzed fish behaviors of body rubbing, swimming speed, fin elections, and contingency checking behaviors appearing in Table 1.

To test the reliability of observation of duration of self-directed behavior before mark test, a blind test was conducted. A set of 25% (2 out of 8 fish) of the video of the self-directed behavior for a week was blindly analyzed (with no information of date or fish name) by a researcher outside our team, and the results were compared to the original data by Kubo. The data set was highly correlated (linear model: R = 0.994, F = 956.64, $p < 0.0001$, $n = 13$, see S1 Data for raw data), which showed the strong reliability of the original data without blind condition.

**Experiment 2: Marking with blue and/or green elastomer.** As reported in "Results and discussion," all of the 8 fish passed the mark test in Experiment 1. To examine whether the pass rate of this fish species was so high because our stimulus was of high ecological relevance, we used blue and green VIE marking, which do not resemble ectoparasites (Fig 1B). Cleaner fish distinguish a variety of color including blue and green [56]. We avoided the colors of red, orange, or yellow, which are different from brown but more or less similar to parasite. VIE marking of blue and green was done subcutaneously on throat as in Experiment 1. We used 6

another fish of the size range of 64 to 74 mm TL. The protocol was similar to Experiment 1, except for sham marking. We did not do any sham marking. Instead, after day 7, subjects were marked with either green or blue VIE during the following night. Two nights later, the mark was removed and the other color injected. The previous mark could be pushed out by gently pushing the skin around the marking with the fingers. After another 2 nights, the second mark was removed, and the standard brown mark was injected. Subject behaviors were recorded for 2 hours before and after the mirror was exposed.

For Experiment 2, if ecological relevance of the mark is important, we expected cleaners to scrape their throat more frequently and to spend more time posing to the mirror when the mark was brown compared to blue and green marks [25]. Each fish was tested in the presence of a mirror in 4 conditions: no mark, blue mark, green mark, and brown mark. The fish swam normally inside the tank in every condition. In Experiment 2, the mark injection, video recordings, and video analyses were done exclusively by Kubo.

In Experiment 2, Kubo and Sogawa quantified throat scraping behavior in 2-hour video recordings in no treatment, color marks in the absent of mirror and color mark in the presence of mirror in all cases of blue, green, and brown mark. After their recording, Kohda checked the video independently and confirmed the accuracy in their counting of mark scrapings (which were absent except for brown marking with mirror). Kubo observed the time (seconds) of position reflecting their throat on mirror in video during 20 minutes in the presence of mirror in 3 color cases.

**Experiment 3: Placing the mark deeper into the fish tissue.** In the previous study, cleaner fish did not exhibit any reactions to the color mark in absence of mirror [25]. However, colleagues criticized that the fish may have felt the subcutaneous injection, which triggered the scraping when they saw the mark in the mirror (on a supposedly different fish) [7,24]. As it is difficult to assess how subjects perceive the subcutaneous injections, we decided to test some fish with a deep injection of the brown VIE into the throat, i.e., where were assumed that the fish will feel some physical stimulus such as pain or itching. The main aim was to test whether the deep injection would make fish scrape their throat even in the absence of a mirror. If they do, the results would show that the visual feedback is not necessary to enhance the sensual feeling in order to elicit scraping. In that case, it would be more difficult to reconcile the proposed visual sensory feedback loop with the absence of scraping without a mirror in subjects with the standard marking. We injected the standard amount ca. 3 mm into the throat of 6 other cleaner fish. This depth of 3 mm reached the border between skin and muscle tissue (Kohda personal observation). The other marking procedures were the same as those in the other experiments. All fish behaved normally the next day. Behaviors of marked fish were video recorded in both the absence and presence of a mirror.

**Experiment 4: Behavior of marked mirror-naive individuals.** In the original experiment on chimpanzees, 2 mirror-naive individuals were marked and exposed to a mirror for 30 minutes. These apparently did not recognize the reflection as self as they did not show any mark-directed behaviors [1]. We conducted a similar experiment by using 9 mirror-naive cleaner fish. We injected brown color on their throat at night and exposed them to a mirror for 2 hours the next morning. Their behaviors were video recorded. If the previous performance in the mark test was based on self-recognition, we predicted that subjects will be less likely to scrape the throat and/or scrape less frequently than the 14 fish tested in Experiments 1 and 2.

**Experiment 5: Behavior of marked mirror-experienced individuals paired with another marked individual.** We conducted this experiment as an additional test on whether seeing a brown mark on the throat of other individuals may cause throat scraping, supposedly because of the visual sensory feedback loop proposed by both de Waal [24] and Gallup and Anderson [7]. We had 3 sets of 2 adjacent aquaria, using in total 6 fish in total. These fish had already passed the

mirror test but the mark became faint. Pairs of subjects were of similar size, i.e., length differed <5 mm. Four days were allowed for the subject pairs to display low levels of aggression [25], and we marked both individuals in a couple with brown VIE during the same night, following standard marking procedure. We also placed an opaque sheet between the tanks. The sheet was removed the next morning and both subjects filmed for 2 hours. If seeing a fish with a mark triggers feedback with own sensations, subjects should scrape their throat in this experiment. If instead individuals need to see the mark on their own image, we expected no throat scraping.

**Experiment 6: Fish responses to moving the mirror.**   We exposed 6 individuals that had passed the mirror test criterion in the previous experiments (Experiment 1) to 2 new situations. First, we transferred the subjects to a new aquarium that contained 2 covered mirrors of $45 \times 28$ cm$^2$ on opposite ends. After an acclimatization phase of 3 days with covered mirrors, we uncovered one of the 2 mirrors for 3 days. On the fourth day, this mirror was covered again, and the second mirror was kept uncovered. Fish reactions to both mirrors were video recorded during the first 2 hours of respective exposure. If cleaner fish use spatial cues to get habituated to the presence of what they perceive as a stranger, the subjects should show aggression toward their reflection in both mirrors. In contrast, if cleaner fish recognize the mirror image as a specific individual (as self or another fish), then they should not show renewed aggression in this experiment.

## Data analyses

In all experiments, fish behaviors in each test were recorded for 2 hours, and these video recordings were used for all behavioral analyses. All data are presented as the mean and SEM.

All statistical analyses were conducted using R 4.1.1 [63]. We used nonparametric statistics throughout the study (i.e., Mann–Whitney U test, Friedman test, and exact Wilcoxon signed-rank test), except for the analyses of swimming speed in Experiment 1 due to the repeated measures from each subject fish and the analyses in Experiment 2 due to the missing data of repeated measures. Swimming speed of subject fish in Experiment 1 ($n = 8$ fish) was compared among the fish with 4 different treatments: no treatment with mirror, sham mark with mirror, brown mark without mirrors, and brown mark with mirror ($n = 5$ times measurements per treatment per individual), using a LMM with individual ID as a random effect. In Experiment 2, we used a Poisson GLMM or a LMM to compare frequency of throat scraping or to compare time staying in posture reflecting the mirror, respectively, among the fish with different color markings (no treatment, blue, green, and brown), with individual ID as a random effect. We established the significance of the fixed factor by means of a likelihood ratio test comparing the full model with a null model. Significance was adjusted to correct for multiple tests using the sequential Bonferroni correction procedures in nonparametric statistics and using Tukey contrasts in LMMs and Poisson GLMMs. Data were considered significant for $p$-value < 0.05.

## Supporting information

**S1 Data. Raw data of the data represented in this manuscript.**
(XLSX)

## Acknowledgments

We are grateful to the members of the Laboratory of Animal Sociology, Osaka City University for their fruitful discussion.

## Author Contributions

**Conceptualization:** Masanori Kohda, Shumpei Sogawa, Alex L. Jordan, Redouan Bshary.

**Data curation:** Shumpei Sogawa, Naoki Kubo, Satoshi Awata, Shun Satoh, Taiga Kobayashi, Akane Fujita.

**Formal analysis:** Naoki Kubo, Satoshi Awata, Shun Satoh.

**Funding acquisition:** Masanori Kohda, Alex L. Jordan, Shun Satoh, Redouan Bshary.

**Investigation:** Shumpei Sogawa, Naoki Kubo, Akane Fujita.

**Methodology:** Masanori Kohda, Redouan Bshary.

**Project administration:** Masanori Kohda.

**Resources:** Masanori Kohda.

**Software:** Taiga Kobayashi.

**Supervision:** Masanori Kohda, Redouan Bshary.

**Writing – original draft:** Masanori Kohda, Redouan Bshary.

**Writing – review & editing:** Shumpei Sogawa, Alex L. Jordan, Satoshi Awata.

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
