## [Editor Report · Decision Letter 0]

8 Apr 2021

Dear Masanori, 

Thank you for submitting your manuscript entitled "Further evidence of mirror self-recognition of the cleaner fish, and the significance of ecologically relevant marks" for consideration as a Research Article by PLOS Biology.

Your manuscript has now been evaluated by the PLOS Biology editorial staff, as well as by an academic editor with relevant expertise, and I'm writing to let you know that we would like to send your submission out for external peer review.

IMPORTANT: We think this would be best considered as an Update Article to your previous PLOS Biology paper (the papers would be linked to each other, and you would pay a lower Article Processing Charge, but it would be a standalone paper). You can read about Update Articles here: https://journals.plos.org/plosbiology/s/what-we-publish#loc-update-article There's no need for any reformatting as your manuscript is already very concise, but please could you change the article type to "Update Article" when you upload your additional metadata (see next paragraph).

Please re-submit your manuscript within two working days, i.e. by Apr 12 2021 11:59PM.

Kind regards,

Roli

Senior Editor

PLOS Biology

---

## [Decision Letter · Decision Letter 1]

28 May 2021

Dear Masanori,

Thank you very much for submitting your manuscript "Further evidence of mirror self-recognition of the cleaner fish, and the significance of ecologically relevant marks" for consideration as a Update Article at PLOS Biology. Your manuscript has been evaluated by the PLOS Biology editors, an Academic Editor with relevant expertise, and by three independent reviewers.

You'll see that all three reviewers are broadly positive about the study and intrigued by your findings. However, they each raise a number of concerns that will need to be addressed before further consideration. In particular, reviewer #2 requests some further analyses of your video footage, and reviewer #3 asks a number of probing questions about your interpretation of the data and of its implications.

In light of the reviews (below), we will not be able to accept the current version of the manuscript, but we would welcome re-submission of a much-revised version that takes into account the reviewers' comments. We cannot make any decision about publication until we have seen the revised manuscript and your response to the reviewers' comments. Your revised manuscript is also likely to be sent for further evaluation by the reviewers.

We expect to receive your revised manuscript within 3 months. 

**IMPORTANT - SUBMITTING YOUR REVISION**

*Re-submission Checklist*

*Published Peer Review*

*PLOS Data Policy*

*Blot and Gel Data Policy*

Sincerely,

Roli

Roland Roberts

Senior Editor

PLOS Biology

rroberts@plos.org

REVIEWERS' COMMENTS:

Reviewer #1:

[identifies himself as Prof. Culum Brown]

This is a very comprehensive study of the MSR test in cleaner fish and addresses many of the potential criticisms put forward by critics of a previous paper. Importantly, the sample size is now substantial. It would be fair to say that the cleaner fish is now one of the most extensively studied animals in the context of this test outside of primates. For me, many of the previous criticism were simply not valid, but the authors of the current paper have addressed them none-the-less. Perhaps the most important outcome of this paper is the notion that the mark must be salient and or ecologically relevant to the species. Fishes make an excellent model system with which to test this hypothesis further. It is quite clear that familiarity with a mirror is also very important so the subjects learn contingencies between their own movements and their reflections.

On the whole, the paper is well presented and easy to follow. In some instances the language is a little awkward. I've tried to correct some of these cases, but it may be beneficial for the type setters to fix some of the grammar. I felt the language was a little conservative in places, I believe the authors have good reason to be more assertive about these findings.

Specific comments:

L41-42: Or it is possible that fish are self-aware. Ie the test is not broken, it really does what we think it does and fish passed.

L107: Surely the fact there was no response to the invisible elastomer counters this argument?

L167-168: Clearly this isn't the case with elastomer. The colour is irrelevant to the final texture.

L185: Ecologically relevant in as much as the attention of the subject is drawn to the mark. But this does not invalidate the test. It does, however, provide some important information about why some animals / individuals fail.

L196: This reminds me of the behaviour Sneddon and others observed when injecting bee venom and or acid into the lips of fish (trout from memory) to study pain responses.

L231: I don't understand why the mark speeds up learning. I understand that prior exposure the mirror enables the fish to realise that the mirror image is not a threat / conspecific and perhaps later that its actually its own reflection.

L242: whether feedback between a visual…

L268: No it may be they are just familiar with the reflected image. Suggest you ref one of the many studies of familiarity effects in fishes.

L321: Worth noting that there are several species of cleaner fish, and other species are known to "peck" at dots with high contrast, presumably because they mimic food. So there is ample opportunity to test the "ecological significance" in a range of fish species.

L379: It is implied in the text that the fish are wild captured, but here you state they are from fish shops. It needs to be clearly stated that fish shops collect them from the wild.

L430: I have to say that I personally have marked 1000s of fish using VIE and I have never once seen them scrap in an attempt to remove it. Ive used every colour on offer. So the notion that the colour of the mark might change how the injection feels to the fish seems remote in the extreme.

L443: "did not touch the work" is an odd phrase.

L447: "as reported…"

L492: "apparently did not…"

L508 " and we marked… following the standard marking"

Reviewer #2:

This is a possibly highly interesting and possibly quite inspiring study. However, while the information on the manipulations done on the experimental subjects as well as the information on the experimental setting is sufficient, there is a serious lack of information on a most crucial part of this study: How, exactly, were the behaviours of the fish observed and rated?

In experiment 1, there were three 'independent' observers. What did they do? Did each of them independently analyse a subset of trials? In such experiments it would be mandatory by today's standards to have a complete and independent rating of the footage by each of at least two observers and subsequent calculation and reporting of some measure for interobserver reliability. Ideally, the raters would not know the aim of the study, for example, they would not be informed about which of the colour conditions was a critical test with an expected increase in scratching, and which of the colour conditions were just controls.

Of course, based on the manuscript it cannot be excluded that the authors have been following such experimental standards. If so, it would be absolutely necessary to report this in detail.

There is another serious shortage in the behavioural data: Findings on two critical indicative behaviours are reported without any frame of reference provided by the overall activity or specific other activities. Overall, the reported behavioural frequencies are quite low. Therefore, it has to become clear from the data that the difference between critical test conditions and various control conditions cannot be caused by a general increase in behavioural activity. Without any information on other behavioural activities an evaluation of this essential point is not possible. Preferably, the reported frequencies of mark-directed behaviours would be expressed as a percentage of overall activity. At least, this measure or a similar measure should be added.

As all experimental sessions have been videotaped, it should be no problem for the authors to obtain data for additional controls (e.g. interobserver reliability) and for behavioural contextualisation within a reasonable amount of time.

Reviewer #3:

[identifies herself as Diana Reiss]

Review of PBIOLOGY-D-21-00901R1- Further evidence of mirror self-recognition of the cleaner fish, and the significance of ecologically relevant marks

The authors state that they conducted the current study to address some of the criticisms and what were suggested as "shortcomings of the Kohda et al 2019 study and they have conducted 6 experiments to do so. In their abstract they summarize their results as follows: "1) 14/14 new individuals scraped their throat when a brown mark had been provisioned, but only in the presence of a mirror, 2) Blue and green colour marks did not elicit scraping, 3) Intentionally injecting the mark deeper beneath the skin reliably elicited spontaneous scraping also in the absence of a mirror, 4) Mirror-naïve individuals injected with a brown mark scraped their throat with lower probability and/or lower frequency compared to mirror-experienced individuals, 5) In contrast to the mirror images, seeing another fish with the same marking did not induce throat scraping, 6) Moving the mirror to another location did not elicit renewed aggression in mirror-experienced individuals." 

I found the results reported on the responses of the cleaner wrasse in this paper and the previous paper by Kohda et al. 2019 very interesting. However, I remain unconvinced by the findings in this report that the wrasse of showing behavior that is equivalent to the results reported in apes, dolphins and elephants. However, the present study has been well-designed to address some of the criticisms and shortcomings of the previous study and indeed the new results are also very interesting and important to help clarify the factors that may contribute to our understanding of why the fish are scraping the marked area of their bodies in the mirror condition more frequently than in the absence of the mirror. In my comments below I will explain the reasons why I remain unconvinced from the results reported that the cleaner wrasse perceive that the image in the mirror is themselves. 

Experiment 1: Replication of mark-test. During the first five days, fish initially exhibited aggression, and contingency-testing behaviors but these decreased and were hardly observed by seven days as in the previous paper (Kohda et al. 2019). 

Lines 406-409: "This latter phase is indicative of subjects having recognized the mirror- reflection as themselves and hence being ready to pass the mark test (Kohda et al. 2019). Thus, in the present study we started to test subjects after one week of mirror presentation, if they exhibited these reactions (Kohda et al. 2019)." One of the main issues I have with this paper is the lack of qualitive and quantitative data and descriptions of the specific behaviors the wrasse exhibit during mirror exposure prior to the mark test. Notably, in past MSR studies with other species (apes, dolphins and elephants) prior to the mark-test, these animals show a variety of self-directed behaviors (not just one behavior). If we consider our use of a mirror, we learn to interpret the information in the mirror based on learning the contingencies of mirror use and as previously described in the MSR literature, mirror naïve apes, dolphins and elephants, show a progression of stages of behavior that are observable reflections of the perceptions (stage 1 social behavior/ mirror exploratory behavior; stage 2: contingency testing (unusual and/or repetitive behaviors); stage 3; self-directed behaviors (and this is a critical stage as the animals seek out the mirror and use it as a tool in viewing parts of their body not-viewable in the absence of a mirror. It is this behavior—that is a first evidence and critical evidence of MSR - not just the touching or response to the mark. The focus on the mark test makes it very difficult to interpret the results as presented. 

Lines 151 -157: There is an error in the following statement that needs to be corrected. The authors state that the "Cleaner wrasse also currently show the highest rate of passing (94% = 17/18; and one failing fish in Kohda et al. 2019) is incorrect and then list all the other studies that have a lower % except the two studies that report a higher %. Specifically, the authors fail to cite the dolphin studies (Reiss & Marino, 2001 in which 2/2 dolphins passed multiple mark tests and Morrison & Reiss, 2018 in which 2/2 dolphins passed the mark test (4/4 =100%). 

Experiment 2: Green and blue marks do not elicit scraping: Only the brown ecologically relevant mark elicited throat scraping and the authors conclude: "This experiment demonstrates that visual information - with or without a potential physical stimulus from the injection - is not enough to elicit throat scraping. Instead, it appears that the visual stimulus needs to be salient and of negative valence. Only an ecologically relevant mark suggesting the presence of an ectoparasite induced throat scraping". Lines 465-469- If ecological relevance of the mark is important, we expected cleaners to scrape their throat more frequently and to spend more time posing to the mirror (assuming a vertical position exposing the mark to the mirror) when the mark was brown compared to blue and green marks (Kohda et al. 2019).

The finding that the fish positioned more frequently to the brown mark is evidence of the ecological relevance and saliency, as predicted by the authors, but as suggested in previous criticisms, the saliency of the brown mark as opposed to the other marks could result in the positioning based on learning the contingencies of mirror use -moving the body in specific ways results in seeing the mark. Therefore, it is critical to have a behavioral record and description of the types of behaviors, contingency testing and self-directed, that the fish exhibited during mirror exposure. Again, focusing on the mark-directed behavior only in the absence of a detailed account of the behavior during mirror exposure makes it difficult to interpret the results and weakens the claim for MSR in the cleaner wrasse. 

This is an interesting result and as the authors suggest it supports the view that the fish are attending to only the ecologically relevant brown marks. One question I have is whether the blue and green marks are visible on the bodies of the wrasse in the mirror refection? Would the wrasse attend to these marks at all in their natural environment? The fish only scraped the marked area in the presence of the mirror and as the authors report, either the visual mark in the mirror or that the combined visual and physical sensation trigger the scraping response. 

Experiment 4: Mirror- naïve individuals injected with a brown mark scraped their throat with lower probability and/or lower frequency compared to mirror-experienced individuals. This seems to support the view that the sensation of the brown mark elicits some scraping but to a lesser degree in the absence of the mirror and that the visual and physical sensation increase scraping. This result is similar to the results reported by Rajala et al 2010 and further supports the view that the combination of the two sensory inputs, visual and physical sensation may focus the animal's attention and lead to understanding the mark is on self. However, it is important to state that this is not required for the apes, dolphins, or elephants in the past studies as the animals as neither the mark nor sham elicited a mark-directed response prior to or in the absence of the mirror. These differences are important in the comparison of results and how MSR is described across species (see Reiss and Morrison (2017). 

Experiment 5. Again, it is hard to interpret and conclude much from the results presented. As I previously mentioned a more complete description of the behavioral responses of the wrasse toward the other fish is needed here. Did the wrasse orient to the marks on the other fish? Did they approach the wall going toward the mark on the other fish? It would be important to mark the fish when together and determine if they try to remove the mark on each other. Furthermore, as in de Waal, 2005, it was demonstrated that individuals respond differently to an individual whose behavior is synchronized with their own than to another whose behavior is not and this should be included in the discussion and the findings are consistent with those findings. However, this experiment does not provide evidence for MSR. 

Experiment 6: Fish responses to moving the mirror: The results showing that the fish did not show aggressive behavior to their mirror image when the mirror was moved to a new position (wall) in the tank was an interesting finding. The authors conclude "The results show that cleaners do not learn a spatial contingency that allows them to eventually stop aggressing their mirror image. Instead, cleaners must learn about their own individual features and selectively stop showing aggression towards these features, independently of where they see them. Note, however, that the lack of aggression by itself does not show that cleaners recognize the mirror image as self. Alternatively, the subjects could have habituated to what they perceive as one specific other individual, and hence show no aggression no matter where they encounter it. Nevertheless, the cleaners clearly outperform rhesus macaques in this particular experiment, as the latter were highly sensitive to the movement of the mirror (Suarez and Gallup 1986)."

I think there is an alternative explanation that should be included here that is more parsimonious and consistent with the results and interpretation of presented by de Waal et al (2005). As the wrasse have had ample experience seeing a fish in the mirror matching their own behavior, when they see this image at another location, it may be perceived as familiar and thus would not be responded to with aggression. From an ecological perspective, individual recognition amongst animals is well documented. I am not familiar with the literature on conspecific recognition in wrasse and it would be very helpful to include more information about individual recognition in wrasse in this paper. 

This finding may also suggest that the difference in movement of the other fish in contrast to a matching movement in their own image may underlie this differential behavior - similar to the results previously reported by de Waal et al (2005) - they state "Capuchins thus seem to recognize their reflection in the mirror as special, and they may not confuse it with an actual conspecific. Possibly, they reach a level of self-other distinction intermediate between seeing their mirror image as other and recognizing it as self." In the discussion section, the authors concur with deWaal et al 2005 that there indeed may be a continuity in self-awareness and the present study and the Koda 2019 study may represent a case for a an intermediary level of self-awareness contingent on mirror exposure.

Discussion: Lines 293- 296 The authors state "However, the results show that the visual input must be the mirror image rather than a marked conspecific. Thus, at the minimum cleaner fish are able to learn that only the mirror image provides contingencies for self, and use this knowledge to scrape a body part with an apparent parasite attached when spotted in the mirror. "

The response itself, while it may appear as evidence - is not a convincing as one might think. As the authors themselves state on Lines 291 -292. "We cannot exclude that there is an intermediate state of itchiness that triggers throat scraping only in combination with a visual input."

Another more parsimonious alternative is that the fish are demonstrating mirror-guided behavior as they have had the opportunity to learn some of the contingencies of mirror use and the ecologically important brown mark has elicited their attention to the mirror thus focusing their attention and enhanced learning. This is similar to the argument that the fish are learning the contingencies of mirror use but they do not show the level of self-recognition as evidenced in the apes, dolphins and elephant studies. The lack of convincing self-directed behavior prior to marking, weakens the authors' argument that the fish are showing equivalent cognitive behavior. I strongly suggest that this point needs be addressed and included in the paper. 

Overall, this is an important paper as it opens up and extends our ideas and discussion regarding studies of MSR and what aspects of behavior need to be included and described to be able to conduct comparative studies across diverse species. 

In summary

As a cognitive scientist involved in conducting MSR studies, I am fascinated and excited about findings reporting continuities in cognitive behaviors across species. Although these are interesting results, I remain unconvinced that the wrasse understand that the image in the mirror is themselves given the scope of the results presented in this paper. There is not sufficient evidence to differentiate mirror guided behavior from self-directed behavior in the mark tests. A more detailed report and video evidence of the contingency and self-directed behaviors prior to the mark-test are needed to be able to confirm this cognitive ability in the wrasse and make the claim for MSR in the wrasse. However, I do think that the authors have presented some very compelling evidence for mirror-guided behavior in the species and this paper is an important contribution to our understanding of comparative cognition and specifically the capacity for MSR. I hope my comments are helpful in strengthening this very interesting contribution.

---

## [Decision Letter · Decision Letter 2]

11 Oct 2021

Dear Masanori,

Thank you for submitting a revised version of your manuscript "Further evidence for the capacity of mirror self-recognition in cleaner fish, and the significance of ecologically relevant marks" for consideration as a Update Article at PLOS Biology. This revised version of your manuscript has been evaluated by the PLOS Biology editors, the Academic Editor and one of the original reviewers (reviewer #2).

In light of the reviews (below), we are pleased to offer you the opportunity to address the remaining points from reviewer #2 in a revised version that we anticipate should not take you very long. We will then assess your revised manuscript and your response to the reviewer's comments and we may consult the reviewer again.

Note that the Academic Editor provided the following comments when I consulted him/her about this concern:

"It must be possible for the authors to give relative frequencies of behavior, i.e. to compare the critical behavior of throat-scraping with the frequency of a few other behaviors in their data and show that *relative* to these other behaviors throat scraping goes up, down, or stays the same. I don't think it will be hard to do, and all we need in the ms is one or two sentences explaining that whatever the claims are for throat-scraping they also hold if we don't measure the behavior by time unit but measure it relative to the rate of other behavior."

We expect to receive your revised manuscript within 1 month.

**IMPORTANT - SUBMITTING YOUR REVISION**

*Resubmission Checklist*

*Published Peer Review*

*PLOS Data Policy*

*Blot and Gel Data Policy*

Best wishes,

Roli

Roland Roberts

Senior Editor

PLOS Biology

rroberts@plos.org

REVIEWERS' COMMENTS:

Reviewer #2:

Dear authors,

due to your very careful revising your manuscript has been improved at many places.

Regarding my own comments to an earlier version of the ms, I appreciate the addition of at least some measure of consistency between observers, although I think that judging of the video footage by at least two independent observers should be a standard procedure nowadays, applied throughout. ("Independent" means independent observations, not necessarily "blind" observations.)

However, you did not address the most crucial point in my last review: Ruling out the likely possibility that increased scores in certain behaviours could have been a side-effect of a marked increase in general behavioural activity. (Aside from some logical reasoning, my scepticism and criticism here is based on experiments I did with fish and mirrors many years ago. Typically, the fish became much more active after getting close to the mirror. As a consequence, a number of behaviours increased in occurrence, including behaviours the fish directed towards themselves. But the proportion of such self-directed behaviours with regard to total behavioural activity did not increase.)

Just to make the point unequivocally clear, a simple example with numbers: Assume the frequency of a certain behaviour, i.e. shaking the head, is normally quite low and the behaviour occurs, on average, every five hours. During an observation period of two hours the measure of occurrence will be zero in most of the cases. Assume now that some factor leads to a fivefold increase in overall behavioural activity. The likelihood for head-shaking will now be about one per hour. Why is it so difficult for you to add general activity data? That could clarify things compellingly.

Without such data, also your point that there was a difference between a brown mark and marks of other colour is not valid. In this case it would be important as well to contextualise the focus data with data on overall activity.

---

## [Decision Letter · Decision Letter 3]

24 Nov 2021

Dear Masanori,

Thank you for submitting a revised version of your manuscript "Further evidence for the capacity of mirror self-recognition in cleaner fish, and the significance of ecologically relevant marks" for consideration as a Update Article at PLOS Biology. This revised version of your manuscript has been evaluated by the PLOS Biology editors, the Academic Editor and one of the original reviewers.

IMPORTANT:

a) I'm afraid that you'll see that reviewer #2 is still not persuaded, and requests further analyses that s/he has specified in their comments. 

b) The Academic Editor suggests that you change the last sentence in your new paragraph (which now reads "Anyway, these observations make certain of the results of the mark test") to the following sentence: "The similarity in travel distance across conditions suggests that the marked increase in throat-scraping in response to a color mark and mirror is not a by-product of overall activity."

c) Regarding reviewer #2's requests, the Academic Editor says "Instead of briefly reporting on general swimming activity, it should be a count of a variety of behavior patterns to show that the relative contribution of throat-scraping stands out among these patterns. Such an analysis may even deserve its own graph or table. It shouldn't be too difficult to add if they still have all the videotapes, and would more conclusively show how behavior changed." We agree with this suggestion to give this additional analysis more prominence, perhaps as a new Figure panel.

In light of the reviews (below), we are pleased to offer you the opportunity to address the remaining points from the reviewer in a revised version that we anticipate should not take you very long. We will then assess your revised manuscript and your response to the reviewers' comments and we may consult the reviewers again.

We expect to receive your revised manuscript within 1 month.

**IMPORTANT - SUBMITTING YOUR REVISION**

*Resubmission Checklist*

*Published Peer Review*

*PLOS Data Policy*

*Blot and Gel Data Policy*

Sincerely,

Roli

Roland Roberts

Senior Editor

PLOS Biology

rroberts@plos.org

REVIEWER'S COMMENTS:

Reviewer #2:

It is much appreciated that the authors now have tried to include measures of overall activity in the analysis. However, I am not convinced by these data. What has been added is very little. A good comparison would be based on behavioural units (frequency of biting, fin erection, or the like). That there are differences in swimming activity between conditions in the data now provided is rather opening the possibility that increased activity did play a role in the behavioural activities in front of the mirror than ruling it out.

Aside from this, the additional data should be presented in a different way. From my point of view these data represent an essential part of the results, not a minor detail of the methods. Thus, data on overall activity should be reported in the results section. Moreover, based on what I consider good scientific standards, the reporting of results should be separated from interpretation. Biased interpretations should be avoided throughout. 

Sentences like " ... the swimming speed in marked fish with mirror was slightly but significantly faster than the fish in no treatment ( = 11.22, P = 0.011), suggesting that fish that find a parasite on throat seem to be in a hurry to remove the harm ..." are an example of a strongly biased look at the findings. I agree with the academic editor with regard to changing the following sentence. However, I think, the data part has to be moved to the result section and any discussion of it to the discussion part of the "results and discussion section". Furthermore, in my opinion this aspect needs a critical discussion that also takes into account what an alternative explanation would mean to the overall conclusions from the study.

---

## [Editor Report · Decision Letter 4]

23 Dec 2021

Dear Masanori,

Thank you for submitting your revised Update Article entitled "Further evidence for the capacity of mirror self-recognition in cleaner fish, and the significance of ecologically relevant marks" for publication in PLOS Biology. The Academic Editor and I have now assessed your responses and revisions.

Based on this assessment, we will probably accept this manuscript for publication, provided you satisfactorily address the following data and other policy-related requests:

IMPORTANT:

a) Please remove the comma from the Title.

b) Please include the ethics committee protocol approval number in the ethics statement within your paper.

c) Please address my Data Policy requests below; specifically, we need you to supply the numerical values underlying Figs 2AB,3AB, 4 and Table 1. Please cite the location of the data clearly in each relevant Fig legend. 

We expect to receive your revised manuscript within two weeks. 

*Published Peer Review History*

*Early Version*

Sincerely,

Roli

Senior Editor,

rroberts@plos.org,

PLOS Biology

ETHICS STATEMENT:

-- Please include the full name of the IACUC/ethics committee that reviewed and approved the animal care and use protocol/permit/project license. Please also include an approval number.

DATA POLICY:

Regardless of the method selected, please ensure that you provide the individual numerical values that underlie the summary data displayed in the following figure panels as they are essential for readers to assess your analysis and to reproduce it: Figs 2AB,3AB, 4 and Table 1. NOTE: the numerical data provided should include all replicates AND the way in which the plotted mean and errors were derived (it should not present only the mean/average values).

DATA NOT SHOWN?

---

## [Editor Report · Decision Letter 5]

5 Jan 2022

Dear Masanori,

On behalf of my colleagues and the Academic Editor, Frans de Waal, I'm pleased to say that we can in principle accept your Update Article "Further evidence for the capacity of mirror self-recognition in cleaner fish and the significance of ecologically relevant marks" for publication in PLOS Biology, provided you address any remaining formatting and reporting issues. These will be detailed in an email that will follow this letter and that you will usually receive within 2-3 business days, during which time no action is required from you. Please note that we will not be able to formally accept your manuscript and schedule it for publication until you have any requested changes.

PRESS: We frequently collaborate with press offices. If your institution or institutions have a press office, please notify them about your upcoming paper at this point, to enable them to help maximise its impact. If the press office is planning to promote your findings, we would be grateful if they could coordinate with biologypress@plos.org. If you have not yet opted out of the early version process, we ask that you notify us immediately of any press plans so that we may do so on your behalf.

Sincerely,

Roli 

Roland G Roberts, PhD 

Senior Editor 

PLOS Biology

rroberts@plos.org